

# Conservation spillover effect of UNESCO World Heritage Sites into surrounding landscapes

Emilia B. Hyland and John E. Quinn

Biology, Furman University, Greenville, SC, United States

## ABSTRACT

Protected areas (PA) are one of the primary tools for conserving and protecting biodiversity, but their goals have evolved overtime beyond nature conservation to include supporting human communities within and adjacent to the PA. UNESCO World Heritage Sites (WHS) offer a unique perspective on the success of PAs as they fall under three categories, cultural, natural, and mixed heritage sites. The nature of these categories encapsulates the inclusion of human communities in the goals of the WHS. To understand the impact and relationship the WHS has with its surrounding landscape, we assessed changes in three indicators, land use and land cover (LULC), human footprint (HF), and forest landscape integrity index (FLII), across three spatial scales, 1, 10, 100 km from the WHS boundary. We found that there is a conservation spillover effect at least within 1 km of the WHS boundary. In this buffer zone, HF was low and FLII was high. FLII was lower and HF was higher at larger spatial scales. The relationship between the WHS and its surrounding landscape is one reason to support the WHS network, however, management of PAs should be more explicit about this relationship as well as relationships between individual PAs.

## INTRODUCTION

Protected areas (PA) are one of the primary tools for conserving and protecting biodiversity. As a conservation tool, over time the goals of PAs and conservation broadly have changed (*Mace, 2014*; *Watson et al., 2014*). Throughout their evolution, the outcomes of PAs have been scrutinized, in part because of changes in their purpose, but also their varied levels of protection and dynamic interactions with the larger landscape (*UNEP-WCMC IUCN, 2022*; *Allan et al., 2017*; *Watson et al., 2014*; *West, Igoe & Brockington, 2006*; *Oldekop et al., 2016*).

One clear shift in the goals of PAs is how their purpose has expanded to both human and non-human communities, within and beyond their formal boundary (*Mace, 2014*; *MacKinnon, Dudley & Sandwith, 2011*; *Andam et al., 2010*). There is evidence that environmental change inside and outside a PA can be attributed to inadequate management plans and increases in human presence as a consequence of formal protection (*Leberger et al., 2020*; *Allan et al., 2017*; *Shirvani, Abdi & Buchroithner, 2020*).

Corresponding author
Emilia B. Hyland,
emiliahyland@gmail.com

For example, PA establishment has been shown to curtail forest loss within the PA boundary, but that forest loss may be displaced to the buffer zones outside the PA (*Barber et al., 2014*; *Ford et al., 2020*). The level of protection designated to a PA may influence environmental quality within the PA and the interactions of the PA outside its boundary. If a PA allows for extraction of natural resources (*e.g.*, logging) within its boundaries, the result may be detrimental to ecosystem function within the border if left unchecked; however, the tradeoff is that local people can still have access to resources that were previously available to them. In this context, social ecological systems research has shown that some PAs designated for sustainable use (*e.g.*, IUCN categories V and VI), experienced more socioeconomic development that translated to empowerment of local people, fewer livelihood costs, and more cultural benefits (*Oldekop et al., 2016*; *Nakamura & Hanazaki, 2017*). *Oldekop et al. (2016)* also positively correlated socioeconomic benefits with conservation successes.

However, how those regulations within the boundary translate to the broader landscape is an important, but not well understood, factor in evaluating the success of PAs. Past research on these relationships has focused primarily within the PA or the immediate vicinity. For example, *Allan et al. (2017)* studied the effects an increase in human pressure and forest loss has on UNESCO World Heritage Site integrity. They, however, like many others (*Jones et al., 2018*; *Leberger et al., 2020*; *Hazen, 2008*) primarily focused on how these indicators affected ecosystems within the PA, but not the landscape around the PA. Moreover, the body of research that has sought to understand the effect of the spillover of PAs, defined as the residual conservation or protection effect of an established PA into the surrounding landscapes, has been at a local scale focusing on single or multiple PAs, and thus makes general patterns more difficult to identify (*e.g.*, *Andam et al., 2010* or *Hazen, 2008*).

To address this gap, our central research question was, does the UNESCO World Heritage Site (WHS) classification of PAs negatively or positively impact the surrounding landscape and to what spatial extent are those effects felt? As a unique subset of PAs, UNESCO World Heritage Sites provide a framework to understand the interactions between human and natural systems within and beyond PA boundaries. Moreover, international organizations from various sectors like the World Wildlife Fund, ICMM, and Globally Important Agricultural Heritage Systems, have commented on their role in conservation by explicitly mentioning preserving UNESCO World Heritage Sites or including them in their mission (*World Wildlife Fund, 2017*; *ICMM, 2023*; *García et al., 2020*). In addition, WHS offer a perspective of the novelty of this classification to study coupled human and environmental systems and contribute to a global demand to understand the evolving roles of PAs (*Allan et al., 2017*; *UNESCO, 2022*; *Hyland, 2021*).

WHS are areas deemed to have universal value for their cultural, natural, or mixed heritage making them worthy of preservation for current and future generations to experience and learn from. The establishment of a heritage site is proposed by that community (*UNESCO, 2022*), suggesting the WHS may be more reflective of interactions between it and the surrounding communities and the potential spillover effects of protection or damage. However, the ongoing relationship with the heritage site and local

people requires continuous effort (*Hazen, 2008*). Therefore, in this article, we will explore the effects a WHS classification has on the human and natural systems at different scales at a global extent.

## METHODS

### Study scale

We evaluated UNESCO World Heritage Sites (WHS) and their buffer zones of different scales at a global extent (Fig. 1). We used multiple spatially continuous global datasets (land use and landcover (LULC), human footprint (HF), and forest landscape integrity index (FLII)) to determine the spatial variability of the effects of the WHS on the surrounding communities at three scales (1, 11, and 100 km). We chose these measurements because they represent different types of indicators as described by *Groves & Game (2016)*; specifically natural, proxy and constructed indicators. Natural indicators reflect a direct measure of the system. A proxy indicator can be used to capture values without a natural scale. Lastly, a constructed indicator will combine multiple measures to reflect a broader concept or idea. In this project, global LULC data is a natural indicator that describes the trends of the landscape (*ESA GlobCover, 2009*). FLII acts as a proxy indicator for the landscape as it measures forest integrity, a proxy of conservation success (*Grantham et al., 2020*). HF is a constructed indicator because it encapsulates different measurements in one scale (*Wildlife Conservation Society—WCS, and Center for International Earth Science Information Network—CIESIN—Columbia University, 2005*). Further details and decision making on the indicators are discussed below (Table S1).

### Data collection

We obtained primary WHS site location and PA data from the UNESCO World Heritage Site and the World Database on Protected Areas (WDPA) data downloads respectively (*UNESCO, 2022*; *UNEP-WCMC IUCN, 2022*). The original WHS data contained 1,154 WHS points and the original WDPA data consisted of 254,526 polygons. We subsetted the sites by selecting polygons where the WHS points and the WDPA polygons intersected using the select by location tool in ArcGIS Pro. This resulted in 390 WHS points and 740 WDPA polygons. We then used the dissolved boundary tool in ArcGIS to dissolve the boundaries of the selected WDPA polygons to have an accurate representation of the protected area to the fullest extent as there were many overlapping polygons from the WDPA dataset. This process resulted in 7,244 individual polygons. To add the WHS attributes back to the WDPA polygons, we spatially joined the dissolved polygons to the 740 original WDPA polygons that were subsetted by location with WHS points. This resulted in 382 WDPA polygons. Since there were 390 WHS points that were within WDPA polygons, we used the spatial join tool in ArcGIS to join the WDPA polygons to many WHS points, as there are some polygons that contain more than one WHS point. This resulted in 390 polygons that included attributes from both WHS and WDPA. This final set of 390 polygons includes WHS from all three WHS categories. Although cultural WHS mainly protect cultural assets, the ones included in this study are associated with a PA and contribute to protecting biodiversity. For instance, the English Lake District in the

A)

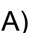

B)

**Figure 1 Global distribution of selected World Heritage Sites (WHS) & primary land use and cover frequency.** (A) Global distribution of selected World Heritage Sites (WHS) that spatially overlapped with WDPA polygons and their buffers. Includes a close up of the English Lake District, Fountains and Studley Royal, and Afon Dyfrdwy WHS to visualize the buffer zones. This area was chosen due to its familiarity to the authors. (B) World primary land use and cover frequency (%) within the three buffers (1, 10, 100 km) and within the World Heritage Site boundary (N = 390). Esri. 2022. https://basemaps.arcgis.com/arcgis/rest/services/World_Basemap_v2/VectorTileServer. (October, 2022). © ESA GlobCover 2022 Project. GlobCover. due ESA. http://due.esrin.esa.int/page_globcover.php.

UK is classified as a cultural WHS, but it contains a significant portion of land and lakes managed for conservation (*UNESCO, 2017*).

Indicators used in the general literature to assess the impact of PAs include human appropriated net primary production (HANPP, *Vačkář et al., 2016*) and forest loss (*Allan et al., 2017*; *Leberger et al., 2020*; *Barber et al., 2014*; *Ford et al., 2020*). To differentiate from these studies, we used global land use and land cover (LULC, *ESA GlobCover, 2009*), human footprint (HF, *Wildlife Conservation Society—WCS, and Center for International Earth Science Information Network—CIESIN—Columbia University, 2005*) and forest landscape integrity index (FLII, *Grantham et al., 2020*) as indicators of human and environmental change associated with the classification of a WHS. HF is measured in global hectares and FLII is measured by index values ranging from 0–10,000. We then downloaded the available indicator datasets from their already established online databases in shapefile or raster format (Table S1). We used ArcGIS Pro 2.8.0 and R v1.4.1717 for further analysis.

## Data processing

We imported the WHS shapefile and indicator rasters into R. We used the st_buffer tool in the sf package (*Pebesma, 2018*) to create buffers of 1, 10, and 100 km around each individual WHS polygon. We selected these three distances for logarithmic scale and to encapsulate different human and natural drivers that might impact change including policy, ecosystems gradients, or land use history (*Ordway et al., 2021*). We used the exact_extract function in the exactextractr package (*Baston, 2022*) to calculate the mean and variance of HF and FLII and mode of the global land use and land cover raster in the different scales and core WHS polygon.

## Data analysis

We observed the frequency distribution of land use and land cover to descriptively compare between scales, regions, and categories. For FLII and HF, we used linear models and AICc model selection (*Burnham & Anderson, 2002*) to determine what combinations of buffer distance, WHS category, and IUCN category had the best support to explain the variation in each indicator. We considered individual models for each variable, combinations of variables, and interactions between variables for FLII and HF. We used the top model ($\Delta$AIC <2) and 95% confidence intervals to understand the direction, magnitude, and confidence in model estimates.

## RESULTS

After filtering WHS with the protected areas database, most of the remaining WHS are in Europe and North America, about 40% or 153 out of 390. The most common IUCN category is Not Applicable, about 28% of WHS or 110 out of 390. Of the WHS that do list their IUCN category, II is the most common with 74 of the 390. The remaining IUCN categories are as follows, V with 43, VI with 23, and IV with 17, Ia with seven, Ib with six, and III with three WHS (Table S2). The most common WHS category type is natural with 197, then cultural with 156, and mixed with 37.

## Descriptive patterns of land use and land cover

The most common LULC in the WHS varied as a function of distance (Fig. 1B), region, WHS category, and IUCN category (Fig. 2). For example, as distance increased, cropland frequencies were greater, and forest vegetation lower. These differences become varied further when considering the effect of the other variables, region, WHS category, and IUCN category with distance on landcover.

Patterns of LULC types across the three scales were different in the five regions. For example, of WHS in the Arab States, inside the border and beyond share bare areas and water bodies LULC types; however, the other types change with buffer distance. Within the PA boundary, mosaic vegetation was the only modal landcover type besides bare areas and water bodies (Fig. 2A). However, this changes to mosaic croplands within 1 km of the WHS border and to only bare areas within 100 km. Moreover, for the WHS in Africa, there was no modal artificial surface and associated areas within or beyond the WHS, whereas in Europe and North America, artificial surfaces were present inside and within 10 km of WHS (Fig. 2A).

The IUCN categories lend themselves to different frequencies of LULC types across spatial scales. For instance, around Ib, one of the stricter conservation IUCN categories, the most frequent LULC type was open needle leaved deciduous or evergreen forest. Interestingly, in this IUCN category, there was higher heterogeneity of LULC within the WHS than in the surrounding landscape (Fig. 2B). In Ia, the most frequent landcover beyond the WHS boundary was artificial surfaces and associated areas 1 and 10 km from the WHS boundary but was not present 100 km from the border (Fig. 2B). These two categories, in addition to category III, have the lowest diversity of LULC types. Category II, the most commonly reported IUCN category, had much more variation. In this category, forest vegetation decreases just beyond the WHS border, but remains relatively consistent across buffer distances. In addition, it is only in the 10 km buffer and beyond that there is a cropland presence. Interestingly, category VI, which allows for sustainable resource use, demonstrated an increasing frequency of forested habitats as buffer distance increased (Fig. 2B). IUCN category Not Applicable was the most common category in this study and has the highest variation of landcover across the buffer distances. It also includes the highest frequency of bare areas and permanent snow and ice.

Similar to the other variables, the three WHS categories illustrated varying patterns in LULC frequencies across the buffer distances (Fig. 2C). One noticeable result is that natural WHS have a consistent frequency of permanent snow and ice across the buffer distances (Fig. 2C). Beside natural WHS, only inside the mixed WHS was there a presence of permanent snow and ice. The four types of croplands also varied over buffer distance in natural WHS (Fig. 2C). Mixed WHS have the highest frequencies of agricultural landcover, especially within 10 and 100 km of the WHS border (Fig. 2C). Another notable pattern is a slight increase in bare areas as buffer distance increases in cultural WHS (Fig. 2C). This may highlight the remote nature of some cultural WHS. Artificial surfaces and associated areas remain at a similar frequency between inside the cultural WHS and 1 km beyond its boundary but is greater 10 km from the boundary. Despite more bare areas and artificial
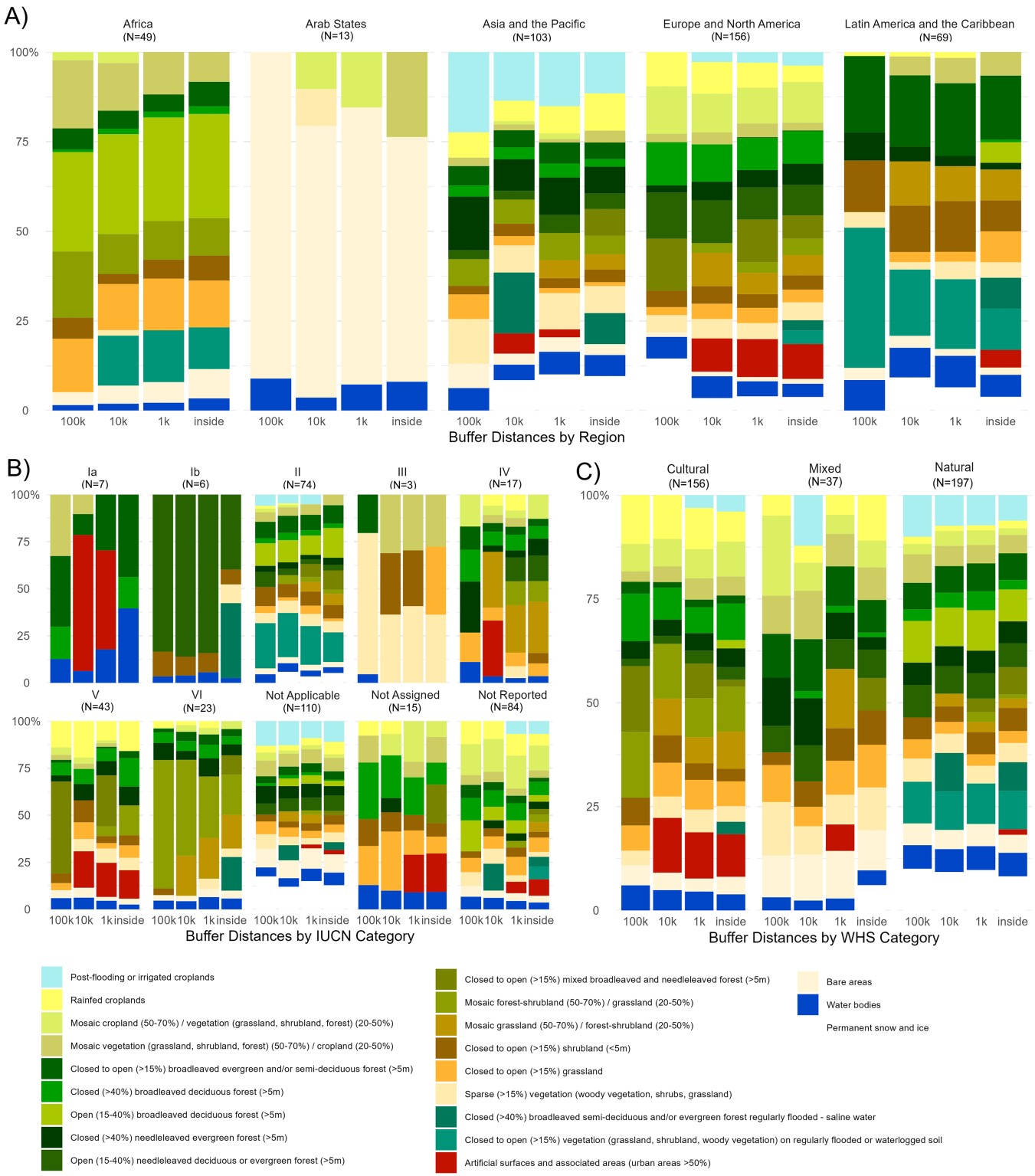

**Figure 2** **Land use and landcover frequencies by region, IUCN category, and World Heritage Site category.** Land use and land cover frequency (%) by (A) biogeographic region, (B) IUCN category, and (C) World Heritage Site (WHS) category across the three scales (1, 10, 100 km) and within the WHS boundary.

**Table 1 AIC model results for human footprint and forest landscape integrity index.**

| FLII | K | AICc | Δ AICc | AICc weight | LL |
|---|---|---|---|---|---|
| Distance + IUCN*WHS | 29 | 15,237.3 | 0.0 | 1.0 | −7,589.1 |
| IUCN*WHS | 26 | 15,257.9 | 20.6 | 0.0 | −7,602.5 |
| Full model | 16 | 15,270.8 | 33.5 | 0.0 | −7,619.2 |
| Distance + IUCN | 14 | 15,282.4 | 45.1 | 0.0 | −7,627.1 |
| Distance + WHS | 7 | 15,301.1 | 63.8 | 0.0 | −7,643.5 |
| IUCN | 11 | 15,301.9 | 64.6 | 0.0 | −7,639.8 |
| WHS | 4 | 15,320.1 | 82.8 | 0.0 | −7,656.0 |
| Distance | 5 | 15,323.3 | 86.0 | 0.0 | −7,656.6 |
| Null | 2 | 15,341.9 | 104.6 | 0.0 | −7,668.9 |
| **HF** | | | | | |
| Distance + IUCN*WHS | 28 | 22,348.9 | 0.0 | 1.0 | −11,145.8 |
| IUCN*WHS | 25 | 22,368.1 | 19.2 | 0.0 | −11,158.5 |
| Full Model | 16 | 22,368.7 | 19.8 | 0.0 | −11,168.1 |
| Distance + WHS | 7 | 22,508.6 | 159.7 | 0.0 | −11,247.2 |
| Distance + IUCN | 14 | 22,518.9 | 170.0 | 0.0 | −11,245.3 |
| WHS | 4 | 22,524.5 | 175.7 | 0.0 | −11,258.3 |
| IUCN | 11 | 22,536.4 | 187.5 | 0.0 | −11,257.1 |
| Distance | 5 | 22,781.1 | 432.2 | 0.0 | −11,385.5 |
| Null | 2 | 22,794.3 | 445.5 | 0.0 | −11,395.2 |

Note:
AIC table for models of human footprint (HF) and forest landscape integrity index (FLII). Distance is the three buffer distances, IUCN represents the IUCN category of the site, and WHS signifies the site's World Heritage Site category. The * symbol indicates an interaction between variables. K represents the number of parameters in the model, LL stands for the log likelihood, AICc is the Akaike information criterion statistical analysis, and delta AICc represents the change in AIC values. AICc weight is the sum of the differences.

surfaces in cultural WHS, the frequency of forested areas is greater. This pattern is also present in mixed WHS; as buffer distance increases, forested area is greater (Fig. 2C).

## Statistical analysis of human footprint and forest landscape integrity index

For both HF and FLII mean and variance, the model that was supported best by the data was buffer distance along with the interaction between IUCN and WHS categories (Table 1). As buffer distance increased, mean HF was higher, and the variance was lower (Table 2, Fig. 3A). The 95% confidence intervals for both the mean and variance estimates were bound by zero, giving strength to the observed pattern. FLII was highest inside the WHS and lower with greater distance of the buffer; variance was lowest inside WHS (Table 2, Fig. 3C). Estimates of the mean were less certain, with only FLII inside the WHS not overlapping zero. Estimates of variance were more certain.

HF for both cultural and natural WHS was highest in IUCN category Ia (Fig. 3B, Table 2) followed by Ib in cultural and V in natural. The former result aligns with the difference seen in the interaction between IUCN category and landcover where Ia had the high artificial and associated land frequencies (Fig. 2B). Ia also has the lowest FLII for cultural WHS but was similar to other IUCN categories in natural sites (Fig. 3D). Ia was
**Table 2 Model estimates for human footprint and forest landscape integrity index.**

**Top model: Distance_IUCN*WHS**

| | Human footprint | | | | | |
| | Mean | | | Variance | | |
| | Estimate | SE | 95% CI | Estimate | SE | 95% CI |
|---|---|---|---|---|---|---|
| (Intercept) | 74.1 | 17.6 | [108.6–39.6] | 472.4 | 394.4 | [1,245.4 to −300.7] |
| Distance10k | **−6.6** | **2.5** | **[−1.7 to −11.5]** | **−415.5** | **56.9** | **[−304.0 to −526.9]** |
| Distance1k | **−8.7** | **2.5** | **[−3.8 to −13.6]** | **−534.1** | **56.9** | **[−422.7 to −645.5]** |
| Distance inside | **−12.8** | **2.5** | **[−7.9 to −17.7]** | **−627.1** | **56.9** | **[−515.6 to −738.5]** |
| IUCN_Ib | 1.7 | 21.5 | [43.8 to −40.4] | 25.0 | 481.2 | [968.1 to −918.1] |
| IUCN_II | −14.4 | 18.0 | [20.9 to −49.7] | 605.3 | 403.1 | [1,395.4 to −184.8] |
| IUCN_III | −40.5 | 21.5 | [1.6 to −82.6] | 251.9 | 481.2 | [1,195.0 to −691.3] |
| IUCN_IV | −3.5 | 18.3 | [32.4 to −39.4] | 479.1 | 410.4 | [1,283.4 to −325.2] |
| IUCN_V | −13.3 | 17.8 | [21.6 to −48.2] | 436.5 | 398.2 | [1,216.9 to −343.9] |
| IUCN_VI | −10.6 | 18.0 | [24.7 to −45.9] | 642.6 | 404.3 | [1,435.0 to −149.8] |
| IUCN_NAP | −13.8 | 19.2 | [23.8 to −51.4] | 755.8 | 430.4 | [1,599.3 to −87.7] |
| IUCN_NA | −16.2 | 18.3 | [19.7 to −52.1] | 417.2 | 410.4 | [1,221.5 to −387.1] |
| IUCN_NR | −15.1 | 17.7 | [19.6 to −49.8] | 440.6 | 397.0 | [1,218.6 to −337.5] |
| Mixed | −30.4 | 18.0 | [4.9 to −65.7] | 149.3 | 404.3 | [941.7 to −643.0] |
| Natural | 12.9 | 18.9 | [49.9 to −24.1] | 40.3 | 424.4 | [872.1 to −791.4] |
| IUCN_Ib:Mixed | | | | | | |
| IUCN_II:Mixed | 4.2 | 20.5 | [44.4 to −36.0] | 45.7 | 458.4 | [944.2 to −852.9] |
| IUCN_III:Mixed | 33.7 | 28.0 | [88.6 to −21.2] | −359.7 | 628.5 | [872.2 to −1,591.5] |
| IUCN_IV:Mixed | | | [0.0–0.0] | | | |
| IUCN_V:Mixed | **104.7** | **25.3** | **[154.3–55.1]** | −664.0 | 567.4 | [448.1 to −1,776.2] |
| IUCN_VI:Mixed | | | | | | |
| IUCN_NAP:Mixed | 8.7 | 20.0 | [47.9 to −30.5] | −764.2 | 448.4 | [114.6 to −1,643.0] |
| IUCN_NA:Mixed | | | | | | |
| IUCN_NR:Mixed | 29.2 | 19.4 | [67.2 to −8.8] | −206.5 | 434.4 | [644.9 to −1,057.9] |
| IUCN_Ib:Natural | **−51.8** | **24.3** | **[−4.2 to −99.4]** | 764.6 | 543.9 | [1,830.7 to −301.5] |
| IUCN_II:Natural | −26.2 | 19.5 | [12.0 to −64.4] | −58.7 | 437.3 | [798.4 to −915.8] |
| IUCN_III:Natural | | | | | | |
| IUCN_IV:Natural | −13.2 | 20.9 | [27.8 to −54.2] | 132.9 | 468.9 | [1,051.8 to −786.1] |
| IUCN_V:Natural | 13.8 | 20.7 | [54.4 to −26.8] | 180.7 | 463.8 | [1,089.8 to −728.4] |
| IUCN_VI:Natural | −31.3 | 20.9 | [9.7 to −72.3] | −570.3 | 469.1 | [349.1 to −1,489.7] |
| IUCN_NAP:Natural | −14.7 | 20.6 | [25.7 to −55.1] | −279.3 | 461.3 | [624.9 to −1,183.5] |
| IUCN_NA:Natural | −6.1 | 21.5 | [36.0 to −48.2] | 556.6 | 482.4 | [1,502.1 to −388.9] |
| IUCN_NR:Natural | −17.0 | 19.4 | [21.0 to −55.0] | 277.6 | 434.3 | [1,128.9 to −573.6] |

| | Forest landscape integrity index | | | | | |
| | Mean | | | Variance | | |
| | Estimate | SE | 95% CI | Estimate | SE | 95% CI |
|---|---|---|---|---|---|---|
| (Intercept) | 640.2 | 1,212.0 | [3,015.7 to −1,735.3] | 3,723,006.0 | 1,421,098.0 | [6,508,358.1–937,653.9] |
| Distance10k | −153.5 | 191.6 | [222.0 to −529.0] | **−2,042,706.0** | **224,697.0** | **[−1,602,299.9 to −2,483,112.1]** |

(Continued)

| | Forest landscape integrity index | | | | | |
| | Mean | | | Variance | | |
| | Estimate | SE | 95% CI | Estimate | SE | 95% CI |
|---|---|---|---|---|---|---|
| Distance1k | −15.4 | 194.5 | [365.8 to −396.6] | **−2,795,311.0** | **228,038.0** | **[−2,348,356.5 to −3,242,265.5]** |
| Distance inside | **749.0** | **194.1** | **[1,129.4–368.6]** | **−3,520,315.0** | **227,592.0** | **[−3,074,234.7 to −3,966,395.3]** |
| IUCN_Ib | **8,962.6** | **1,706.0** | **[12,306.4–5,618.8]** | −1,039,815.0 | 2,000,424.0 | [2,881,016.0 to −4,960,646.0] |
| IUCN_II | **5,562.3** | **1,246.6** | **[8,005.6–3,119.0]** | 1,177,557.0 | 1,461,694.0 | [4,042,477.2 to −1,687,363.2] |
| IUCN_III | **5,580.7** | **1,618.6** | **[8,753.2–2,408.2]** | 2,243,324.0 | 1,897,965.0 | [5,963,335.4 to −1,476,687.4] |
| IUCN_IV | **3,150.7** | **1,266.7** | **[5,633.4–668.0]** | 2,183,061.0 | 1,485,298.0 | [5,094,245.1 to −728,123.1] |
| IUCN_V | **2,887.0** | **1,224.5** | **[5,287.0–487.0]** | 2,008,989.0 | 1,435,791.0 | [4,823,139.4 to −805,161.4] |
| IUCN_VI | **5,708.0** | **1,252.8** | **[8,163.5–3,252.5]** | 2,861,864.0 | 1,468,959.0 | [5,741,023.6 to −17,295.6] |
| IUCN_NAP | **3,620.6** | **1,379.6** | **[6,324.6–916.6]** | 2,506,121.0 | 1,617,693.0 | [5,676,799.3 to −664,557.3] |
| IUCN_NA | 2,163.8 | 1,275.4 | [4,663.6 to −336.0 | 2,534,087.0 | 1,495,444.0 | [5,465,157.2 to −396,983.2] |
| IUCN_NR | **2,513.6** | **1,221.1** | **[4,907.0–120.2]** | 2,129,268.0 | 1,431,781.0 | [4,935,558.8 to −677,022.8] |
| Mixed | 2,162.1 | 1,252.8 | [4,617.6 to −293.4] | −634,947.0 | 1,468,959.0 | [2,244,212.6 to −3,514,106.6] |
| Natural | **6,746.9** | **1,348.7** | **[9,390.4–4,103.4]** | 2,245,912.0 | 1,581,474.0 | [5,345,601.0 to −853,777.0] |
| IUCN_Ib:Mixed | | | | | | |
| IUCN_II:Mixed | −778.5 | 1,425.4 | [2,015.3 to −3,572.3] | 1,959,330.0 | 1,671,434.0 | [5,235,340.6 to −1,316,680.6] |
| IUCN_III:Mixed | −379.0 | 2,046.7 | [3,632.5 to −4,390.5] | −743,325.0 | 2,399,932.0 | [3,960,541.7 to −5,447,191.7] |
| IUCN_IV:Mixed | | | | | | |
| IUCN_V:Mixed | | | | | | |
| IUCN_VI:Mixed | | | | | | |
| IUCN_NAP:Mixed | −105.4 | 1,452.5 | [2,741.5 to −2,952.3] | 47,459.0 | 1,703,157.0 | [3,385,646.7 to −3,290,728.7] |
| IUCN_NA:Mixed | | | | | | |
| IUCN_NR:Mixed | 1,116.3 | 1,349.3 | [3,760.9 to −1,528.3] | **3,224,384.0** | **1,582,191.0** | **[6,325,478.4–123,289.6]** |
| IUCN_Ib:Natural | **−8,210.4** | **1,907.4** | **[−4,471.9 to −11,948.9]** | 1,812,762.0 | 2,236,542.0 | [6,196,384.3 to −2,570,860.3] |
| IUCN_II:Natural | **−5,717.7** | **1,397.0** | **[−2,979.6 to −8,455.8]** | −775,667.0 | 1,638,071.0 | [2,434,952.2 to −3,986,286.2] |
| IUCN_III:Natural | | | | | | |
| IUCN_IV:Natural | **−3,520.3** | **1,486.9** | **[−606.0 to −6,434.6]** | −2,499,200.0 | 1,743,501.0 | [918,062.0 to −5,916,462.0] |
| IUCN_V:Natural | **−3,215.7** | **1,492.3** | **[−290.8 to −6,140.6]** | −1,676,844.0 | 1,749,834.0 | [1,752,830.6 to −5,106,518.6] |
| IUCN_VI:Natural | **−5,563.2** | **1,491.4** | **[−2,640.1 to −8,486.3]** | **−4,845,717.0** | **1,748,777.0** | **[−1,418,114.1 to −8,273,319.9]** |
| IUCN_NAP:Natural | **−4,360.8** | **1,514.7** | **[−1,392.0 to −7,329.6]** | −2,654,401.0 | 1,776,102.0 | [826,758.9 to −6,135,560.9] |
| IUCN_NA:Natural | **−4,189.6** | **1,534.3** | **[−1,182.4 to −7,196.8]** | −764,930.0 | 1,799,105.0 | [2,761,315.8 to −4,291,175.8] |
| IUCN_NR:Natural | **−4,283.8** | **1,382.9** | **[−1,573.3 to −6,994.3]** | −1,570,768.0 | 1,621,562.0 | [1,607,493.5 to −4,749,029.5] |

**Note:**
Model estimates and standard error for the top model Human Footprint (HF) and Forest Landscape Integrity Index (FLII) mean and variance. Interactions between WHS and IUCN categories are noted with a ":". Estimates with 95 % confidence intervals (95% CI) not overlapping zero are noted in bold. SE stands for standard error. IUCN_NAP is for not applicable, IUCN_NA is for not assigned, and IUCN_ NR for not reported IUCN categories.

not present for mixed WHS for either HF or FLII. The lowest HF in natural sites fell under IUCN category Ib, which also had the highest FLII (Figs. 3B and 3C), though there was high variance in Ib for cultural WHS. FLII was high in IUCN category VI in all three WHS

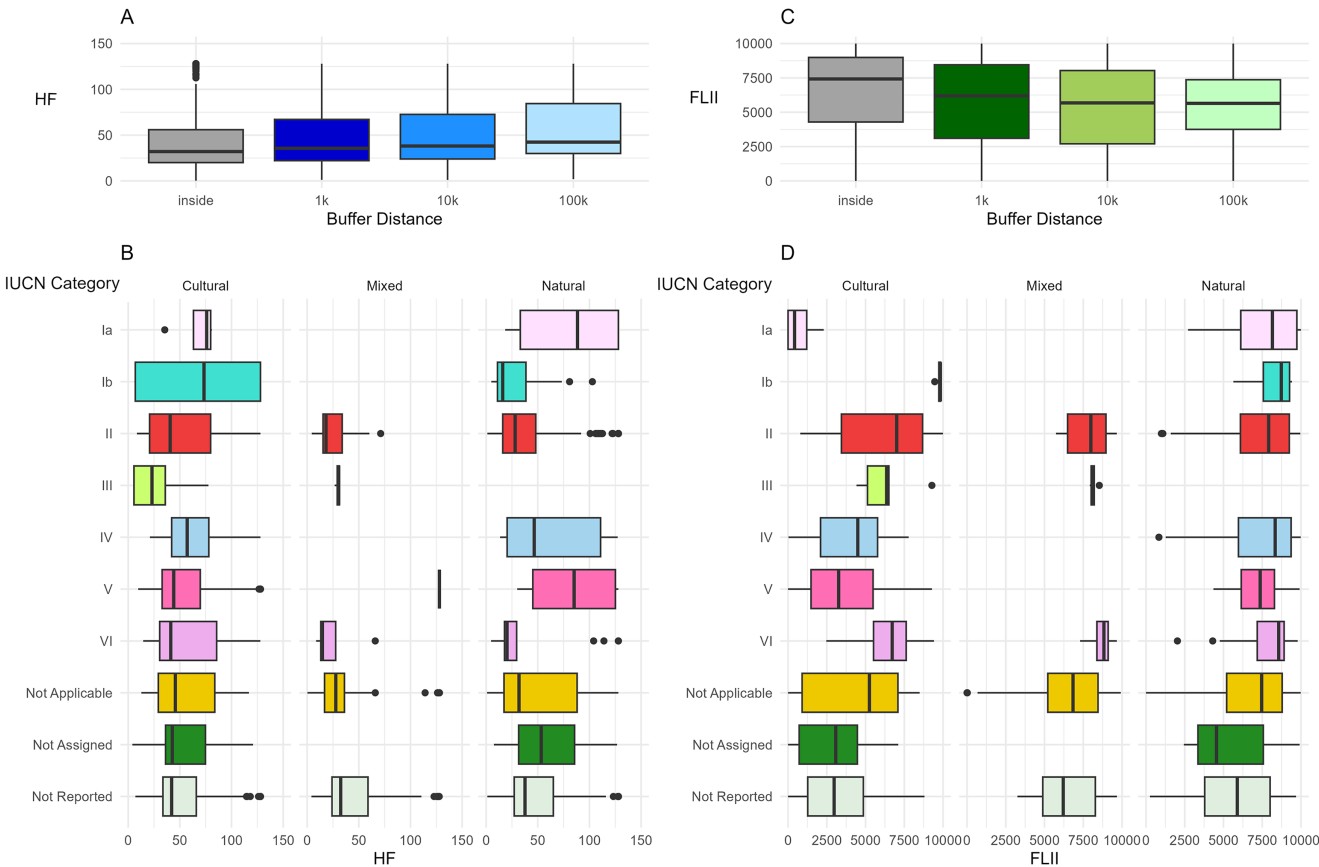

**Figure 3** Human footprint and forest landscape integrity index by distance, IUCN category (IUCN), and World Heritage Site (WHS) category. Variation in (A) human footprint (HF) and (C) forest landscape integrity index (FLII) as a function of distance and the interaction between IUCN and WHS categories for (B) HF and (D) FLII.

categories and had high variance in each Not Applicable category. The former result also aligns well with the IUCN and landcover results as VI had the highest frequency of mosaic forest-shrubland across the three buffer distances (Fig. 2B). Additionally, IUCN type II had high FLII values across WHS categories. Though there were fewer mixed WHS, across the IUCN categories, HF is low and FLII is high apart from a small number of outliers in each.

## DISCUSSION

Our analysis illustrates our understanding of the evolution of PAs to include interactions between humans and nature at the global scale. Exploring these relationships is key to conservation success and development management plans that consider the impacts of decisions within and beyond the PA boundary and guide resource allocation and access (*Li, Wu & Cai, 2008*; *Quinn & Wood, 2017*; *Al-Tokhais & Thapa, 2019*). WHS offer an opportunity to understand this relationship and its effects at a global extent.

The results of our analysis provide evidence of a spatial relationship between WHS and the surrounding landscape and suggest the potential positive effect the WHS classification can have on the surrounding landscape. For instance, there was a clear increase in HF at greater spatial extents and a decline in FLII. However, there was great variation in this

pattern amongst the interactions between IUCN and WHS categories. For example, IUCN category VI had one of the highest FLII medians in all three WHS categories, suggesting better conservation success with sustainable use of natural resources within the site. Likewise, the descriptive patterns in LULC highlight how different conservation classifications and regions could influence land use within and beyond the WHS boundary, particularly when comparing the most common landcover experienced in those extents. For example, the surrounding landscapes of WHS categorized as Ia and VI have different compositions, highlighting the potential effect of strict *vs* sustainable regulations. In this instance, the PA classification may be curtailing human pressures within the WHS but could be offsetting that development directly outside the border. Our global perspective also adds nuance to the discussion as land use and land cover patterns are heavily influenced by the biogeographical location of the WHS including climate and historical land use.

## Future research

The observed patterns in LULC associated with the WHS suggest testable relationships for future research. For example, a temporal analysis could test if the pattern of increasing cropland landcover beyond natural WHS is a consequence of lack of development within many natural WHS or if agriculture in the region has been misplaced due to WHS/PA establishment. This dichotomy is important to explore given the evidence that WHS/PA can serve as sources for food for local people (*Nakamura & Hanazaki, 2017*; *Oldekop et al., 2016*). A lack of consideration of these interactions can result in increasing tensions between the PA and local communities and translate into poor conservation success.

In our subset of WHS there are more natural WHS than there are cultural WHS, the inverse of the full WHS network. The different needs for cultural WHS than for natural suggest interesting questions. *Comer (2012)* states that preservation should be emphasized for cultural/archeological sites rather than conservation in natural sites because cultural sites are not able to bounce back like natural sites. Therefore, the impacts of tourism from the WHS label may be more drastic on cultural sites than natural. Indeed, in our data, though uncertain, HF was higher in the natural than cultural (and lowest in the mixed) and FLII was highest in mixed.

While there is clear value to a global analysis, it is still important to also consider the individual relationships a WHS has with the boarding landscape as these results may not be true for each WHS. In a case study analysis, the positive conservation spillover effect was not consistent at each site (*Hyland, 2021*). For instance, FLII decreased steadily as distance from WHS boundary increased in the Río Plátano Biosphere Reserve, whereas, in the Great Smoky Mountains, FLII dropped drastically 1 km from boundary, but then increased again in the 10 km buffer zone (*Hyland, 2021*). The decrease of FLII in the 100 km buffer zone and increase in HF in the 100 km buffer zone may point to a larger problem at hand (*Hyland, 2021*; *Radeloff et al., 2010*). The isolation of PAs globally. This isolation makes it difficult for species to travel resulting in less genetic diversity, more fatalities, and overall habitat loss (*Fahrig, 2003*). This will be especially felt with climate

change as species ranges shift in response to changing temperatures and precipitation while the PAs will remain constant (*Rannow et al., 2014*; *Elsen et al., 2020*).

## Study limitations

Because of the global nature of study, there were limitations in available datasets for indicators. High quality, fine grain, and easily accessible global data is lacking for environmental indicators. For instance, we had completed the analysis for human appropriated net primary production (HANPP), but then later decided to remove it from the study because the grain size was much larger than the other indicators relative to the size of our buffers. Even though it was removed from the study, HANPP showed similar patterns to FLII and HF with increasing averages as distance increased from the WHS boundary.

In addition, better reporting on IUCN categories should be a priority for the UNESCO WHS convention for current and future WHS as the most common type in this study is Not Applicable. These categories are crucial to highlighting trends in the relationship between protection and environmental integrity of the PA itself and its neighboring landscape. These patterns may be the key to highlighting and implementing effective conservation methods and efforts of these culturally and naturally significant sites and their surrounding landscapes.

## CONCLUSION

The relationship between a PA and the bordering landscape is important for conservation success within the boundary and environmental integrity beyond. WHS, a global example of the evolution of the purpose of PAs, can have a positive conservation spillover effect into the surrounding landscape at least at the 1 km scale from the PA boundary. The scale of this effect is influenced by the distance from the WHS boundary and IUCN and WHS category type. However, studying the spatial and developmental context of each WHS before inscription is still needed to fully support conservation efforts.

Understanding how PAs interact with and affect the surrounding landscapes is crucial to enhancing conservation efforts and meeting conservation goals globally. Landscape change outside the PA boundary can affect conservation methods and effectiveness within the PA itself and vice versa (*Fuller et al., 2019*). The spatial relationship between PAs and local communities can create tensions that limit the PA's conservation success (*West, Igoe & Brockington, 2006*; *Al-Tokhais & Thapa, 2019*; *Nakamura & Hanazaki, 2017*). Tensions can be the displacement of local people after the creation of a PA and loss of access to once available resources (*West, Igoe & Brockington, 2006*; *Wittemyer et al., 2008*; *Mombeshora & Le Bel, 2009*). Acknowledging these tensions is important because the people who are directly influenced by the designation of a PA are crucial to the success of its management plan. If they are removed from the land, they are often left out of the conversation (*Al-Tokhais & Thapa, 2019*). Unsuccessful management plans lack support by the local community, and they are often undermined by illegal activity (*Wiesmann, Liechti & Rist, 2005*; *Al-Tokhais & Thapa, 2019*; *Brandon & Wells, 1992*; *Mombeshora & Le Bel, 2009*). Management plans, however, can be weakened in several other ways too. They may not

receive enough funding, legal protection, or staff to enact and maintain the management plan (*Li, Wu & Cai, 2008*; *Hazen, 2008*; *Watson et al., 2014*; *Oldekop et al., 2016*). The effectiveness of PAs, therefore, comes down to adequate funding, local participation, and in general, a good relationship with locals. The results from this study further illustrate the importance of a positive relationship with surrounding communities and landscapes for continued conservation successes within the WHS boundaries globally. These findings can be listed as a potential benefit to listing a location as a WHS in the future.

## ACKNOWLEDGEMENTS

We thank Furman University for supporting this research.

### Funding

The authors received no funding for this work.

### Competing Interests

The authors declare that they have no competing interests.

### Author Contributions

- Emilia B. Hyland conceived and designed the experiments, performed the experiments, analyzed the data, prepared figures and/or tables, authored or reviewed drafts of the article, and approved the final draft.
- John E. Quinn conceived and designed the experiments, performed the experiments, analyzed the data, prepared figures and/or tables, authored or reviewed drafts of the article, and approved the final draft.

### Data Availability

Data are available at the United Nations Educational, Scientific and Cultural Organization, World Database on Protected Areas, European Space Agency, Wildlife Conservation Society, and Grantham (2020):

- UNESCO WHS points: UNESCO. (2022). World Heritage Interactive Map. UNESCO World Heritage Convention. Select XLS from the The World Heritage List (in other formats), select XLS to the right of the "World Heritage List" heading: https://whc.unesco.org/en/interactive-map/.

- WDPA polygons: UNEP-WCMC, IUCN (2022). Protected Planet: The World Database on Protected Areas (WDPA). https://www.protectedplanet.net/en/thematic-areas/wdpa?tab=WDPA.

- Land cover data: ESA GlobCover, 2009 Project. GlobCover. due ESA. (Globcover2009_V2.3_Global_.zip) http://due.esrin.esa.int/page_globcover.php.

- Human footprint data: Wildlife Conservation Society—WCS, and Center for International Earth Science Information Network—CIESIN— Columbia University. 2005. Last of the Wild Project, Version 2, 2005 (LWP-2): Global Human Footprint Dataset

(Geographic). Palisades, New York: NASA Socioeconomic Data and Applications Center (SEDAC). https://doi.org/10.7927/H4M61H5F. Accessed 21 October 2021.

- FLII data: Grantham, H.S., Duncan, A., Evans, T.D., Jones, K. R., Beyer, H.L., Schuster, R., Walston, J., Ray J. C., Robinson, J. G., Callow, M., Clements, T., Costa, H. M., DeGemmis, A., Elsen, P. R., Ervin, J., Franco, P., Goldman, E., Goetz, S., Hansen, A., Hofsvang, E., Jantz, P., Jupiter, S., Kang, A., Langhammer, P., Laurance, W. F., Lieberman, S., Linkie, M., Malhi, Y., Maxwell, S., Mendez, M., Mittermeier, R., Murray, N. J., Possingham, H., Radachowsky, J., Saatchi, S., Samper, C., Silverman, J., Shapiro, A., Strassburg, B., Stevens, T., Stokes, E., Taylor, R., Tear, T., Tizard, R., Venter, O., Visconti, P., Wang, S., & Watson, J. E. M. (2020). Anthropogenic modification of forests means only 40% of remaining forests have high ecosystem integrity. Nature Communication 11, 5978. https://doi.org/10.1038/s41467-020-19493-3.

## Supplemental Information

Supplemental information for this article can be found online at http://dx.doi.org/10.7717/peerj.15858#supplemental-information.

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
