# Peer review of "Conservation spillover effect of UNESCO World Heritage Sites into surrounding landscapes"

_PeerJ, doi:10.7717/peerj.15858_

## Round 0.1 · original submission · Major Revisions

The reviewers and I recognize the value of your manuscript but found a few important issues that need to be addressed. I encourage you to address the issues raised by the reviewers and resubmit this important contribution.

Additionally:

1) Please mention that international organizations like the ICMM have laid out sustainability principles explicitly mentioning World Heritage Sites: https://www.icmm.com/en-gb/our-principles/mining-principles/principle-7

2) Please analyze variance in addition to averages, as suggested by Reviewer 2.

3) Please provide supplementary tables describing all the variables and the details of the spatial layers (source url, resolution, etc).

4) Please provide all the R code needed to replicate all analyses, ideally in an open-access repository (like GitHub).

Reviewer 1 ·

Basic reporting

The manuscript is well written. However, check for small instances e.g. spillover is sometimes it is spelt with a “-“ and others time not and this needed to be standardised.

Please also better define spillover.

Experimental design

While I found the manuscript poses interesting questions, but I do not think the mythology is detailed enough and I think huge assumptions are made which are not explained well. I worry this would affect the findings of your work.

Cultural WSHs, while can be in natural settings, are focused on conserving cultural assets and therefore are often buildings etc in highly urban areas. Therefore they would have no value for the conservation of biodiversity and natural environments around them, as this is not the point. The study seems to focus on the more natural areas in the way it is contextulaised, therefore I really believe it should only focus on natural WHS and maybe also include mixed WHS.

How did you choose the buffer zone size? Please justify this more. A 100 km buffer is huge and in many instances and I am not sure that any meaningful conclusions can be drawn from this. I.e. a 100 km buffer is the size of some European countries and islands which have WHS. I think this may need to be reduced or at least maybe made to be proportional according to the size of the WHS, the country, the biome or common land use in the area.

Linking to your forest landscape index variable, I believe there is again may be a bit of bias created by using this. Many WHS are not in forest biome areas and so this metric is not relevant in many cases.

Line 89-93. Please explain this in a lot more detail and justify their use.

Validity of the findings

I worry a bit about the validity base don the buffer zone size and the core variables used (e.g. forest landscape integrity index)

Reviewer 2 ·

Basic reporting

Several citations are lacking to support author' statements. Tables and figures must be improved and merged as they are too many. Detailed comments are provided below and in the attached PDF to ease this task.

Detail acronyms and symbols when firstly used in the text (e.g. LULC) and again in the figures, as they are often seen by readers independently of the main text. Again, it is not clear if you use Distance as a separate variable as you do not explain in the caption of table 2 the meaning of ‘*’ or ’ _’.

Some sentences in the results are explaining them, as thus should be moved to the discussion (e.g.: line 159-160).

Figure 1.
The authors made a good effort to present data at a global level. However, there are a lot of UNESCO World Heritage sites missing in Figure 1 (e.g.: Hegra Archaeological Site), and presumably in the analyses. Please check the link below and update your figure and analyses accordingly:
https://whc.unesco.org/en/interactive-map/https://whc.unesco.org/en/interactive-map/

I also cannot understand how the 1km and 100 Km buffers are visible in some areas, but not the 5-km. Are you including the site itself within the 1-km buffer? Please clarify this.
Finally, please justify the rationale to zoom up WHS sites in UK instead of another world region.
Figure 2.
Merge with figure 1 with the subtitle (e.g., World; N= 382). Detail acronyms in the figures, as they are often seen by readers independently of the main text.
Figure 3.
Improve the reading of the subtitle on the bottom left corner graph. Include the sample size within each region (number of WHS) below each subtitle.
Figure 4.
Merge with figure 5.
Figure 5.
Check comments above.
Figure 6.
The font size of the captions of the axis and legend is too small. Reduce the width of the top graph to merge it with the top graph of figure 7.
Figure 7
Merge it with figure 6. Move the bottom graph to the bottom of figure 6.
Table 1.
Move it to supplementary material. Percentage symbols should be removed from the rows and included in the heading of the last column. All figures should be justified to the right to improve the reading of the magnitude of the values.
Table 2.
Detail in the caption the meaning of the codes used in the headings and first column. All figures should be justified to the right to improve the reading of the magnitude of the values. Mark the most significant values.
Table 3
Idem. Additionally, format the hight of the rows.
Table 4.
Merge with Table 3 by adding extra columns to the right.

Experimental design

This study is based on a interesting idea of relating land cover, proximity to protected areas and world heritage sites. However, methods are still not described with sufficient detail and information to be replicated. I suggest that you improve the description of the variables used to determine the spatial variability of the effects of the WHS on the surrounding communities. You should include a supplementary table explaining their source, resolution and provide support from previous works for their choice. The resolution of the shapefiles of the Protected Areas and WHS is also lacking. In some parts, it is not clear what you mean. Please clarify what you mean with “social ecological processes” in line 119-120.
You should also provide a short explanation of the statistical analyses performed and how to interpret the results. It seems that you have compared the average values of HF and FLII with the variables characterizing each WHS analyses, but average may not be the best way to compare areas with very different sizes where those values can vary widely within the same area. It would be important to analyse the variation as well, deviation as well. In fact, that is visible in the whisker-boxes you present in Figure 6 and 7 (e.g., Ib IUCN category has one of the highest HF but also the widest variance in values). It would be important to analyse the standard deviation as well.

Validity of the findings

In the results, it is unclear if you are referring to a significant variation. Please clarify and present the statistical analyses supporting those statements. More importantly, the description of the results is bias to the negative effects of distance to PA, as authors omitted some positive effects in the increase of frequency of more pristine habitats with distance (e.g., increase of frequency of forested habitats from inside to 100k-distance in cultural and mixed WHS and in VI IUCN categories; same with artificial areas inside or 100-K in cultural sites). Thus authors have to tone down the conclusion that PAs may curtailing human pressures within WHS as statistical support is lacking and many cases are against this (see comments above) and move those conclusions to the discussion. In sum, as it is the interaction between categories of the IUCN and WHS that is important to explain the results, the interpretation of the effect of distance in those categories on their own is complicated. That is why authors should move the graphs presented in figure 5 and 6 to supplementary material and redraw a figure depicting the interaction between those as a main result.

Additional comments

Please check the detail comments in the attached PDF. I sincerely hope that these comments will help you to improve this manuscript.

Annotated reviews are not available for download in order to protect the identity of reviewers who chose to remain anonymous.

---

## Round 0.2 · Minor Revisions

I thank you for your revised manuscript. Please address the remaining issues raised by the Reviewer.

Additionally, please address the following:

- In Table 2, include 95% confidence intervals to assess which effects are significantly different from zero.

- In your abstract you mention "We found that there is a conservation spillover effect at least within 1 km of the WHS boundary." However, I was unable to find this result. Please include a table or effects figure clearly showing which effects are significant (do not contain zero).

- If you indeed found an effect at 1 km, why didn`t you look at other distances between 1-10 km? I would like to see where the effects stop being significant at a finer spatial scale (maybe 1 km?)

- Please provide a color legend in all color figures.

Reviewer 2 ·

Basic reporting

The manuscript was significantly improved, and the effort of authors was clear, as they have redone their statistical analyses, clarified the methods, and improved the graphs and tables. They have also included the citations and references lacking to support their statements in the introduction and discussion and toned down their statements about the positive effects of the proximity to PAs. This improved the manuscript quality substantially. However, some minor changes should be made to the figures and tables to improve reading. Please check the attached PDF for details.

Experimental design

no comment

Validity of the findings

no comment

Additional comments

no comment

Annotated reviews are not available for download in order to protect the identity of reviewers who chose to remain anonymous.

---

## Round 0.3 · Minor Revisions

Thank you for your much improved manuscript. Please consider the final minor comments made by the Reviewer.

Reviewer 2 ·

Basic reporting

The figures and tables were significantly improved, and the effort of authors was clear. However, some minor changes are still lacking and should be made to improve reading. Please check the attached PDF for details.

Experimental design

no comment

Validity of the findings

no comment

Additional comments

no comment

Annotated reviews are not available for download in order to protect the identity of reviewers who chose to remain anonymous.

---

## Round 0.4 · accepted · Accept

Thank you for addressing all the issues raised by the Reviewers and me over the past months. I believe this is an important and timely contribution, and I am happy to accept your manuscript.